# Oligomycins inhibit *Magnaporthe oryzae Triticum* and suppress wheat blast disease

**Moutoshi Chakraborty**[ID]*, **Nur Uddin Mahmud**[ID]*, **Abu Naim Md. Muzahid, S. M. Fajle Rabby, Tofazzal Islam**[ID]*

Institute of Biotechnology and Genetic Engineering, Bangabandhu Sheikh Mujibur Rahman Agricultural University, Gazipur, Bangladesh

☯ These authors contributed equally to this work.
* tofazzalislam@yahoo.com

**Data Availability Statement:** All relevant data are within the manuscript and Supporting Information files.

**Funding:** This work was funded by the Krishi Gobeshona Foundation (KGF), Bangladesh through

## Abstract

Oligomycins are macrolide antibiotics, produced by *Streptomyces* spp. that show antagonistic effects against several microorganisms such as bacteria, fungi, nematodes and the oomycete *Plasmopara viticola*. Conidiogenesis, germination of conidia and formation of appressoria are determining factors pertaining to pathogenicity and successful diseases cycles of filamentous fungal phytopathogens. The goal of this research was to evaluate the in vitro suppressive effects of two oligomycins, oligomycin B and F along with a commercial fungicide Nativo® 75WG on hyphal growth, conidiogenesis, conidial germination, and appressorial formation of the wheat blast fungus, *Magnaporthe oryzae Triticum* (MoT) pathotype. We also determined the efficacy of these two oligomycins and the fungicide product *in vivo* in suppressing wheat blast with a detached leaf assay. Both oligomycins suppressed the growth of MoT mycelium in a dose dependent manner. Between the two natural products, oligomycin F provided higher inhibition of MoT hyphal growth compared to oligomycin B with a minimum inhibitory concentration of 0.005 and 0.05 µg/disk, respectively. The application of the compounds completely halted conidial formation of the MoT mycelium in agar medium. Further bioassays showed that these compounds significantly inhibited MoT conidia germination and induced lysis. The compounds also caused abnormal germ tube formation and suppressed appressorial formation of germinated spores. Interestingly, the application of these macrolides significantly inhibited wheat blast on detached leaves of wheat. This is the first report on the inhibition of mycelial growth, conidiogenesis, germination of conidia, deleterious morphological changes in germinated conidia, and suppression of blast disease of wheat by oligomycins from *Streptomyces* spp. Further study is needed to unravel the precise mode of action of these natural compounds and consider them as biopesticides for controlling wheat blast.

## Introduction

Oligomycins are macrolide antibiotics, produced by some strains of *Streptomyces*. They have broad-spectrum biological activities against organisms like fungi, bacteria, nematodes and the

a coordinated project No. KGF TF 50-C/17 to
Tofazzal Islam of the Institute of Biotechnology and
Genetic Engineering of BSMRAU, Bangladesh.

**Competing interests:** The authors declare that
there is no conflict of interest.

oomycete *Plasmopara viticola* [1–4]. *Streptomyces* species are common soil-dwelling bacteria
that have been broadly used as bio-control agents [5]. *Streptomyces* species produce a number
of bioactive compounds possessing antifungal, antiviral, antibacterial, anticancer, nematicidal,
and antioxidant properties [5, 6]. Several previous studies showed that the effectiveness of
some strains of *Streptomyces* in biological control of phytopathogens largely depends on the
production of oligomycins [7]. The oligomycins are mitochondrial F1F0 ATP synthase inhibi-
tors that cause apoptosis in a number of cell types [8]. The oligomycin complex, which was
first documented in 1954 in a strain of a soil bacterium, *Streptomyces diastatochromogenes* was
highly inhibitory against fungi [1]. Antifungal, antitumor, insecticidal, immunosuppressive
and nematicidal properties of oligomycins have also been reported [1–3, 7, 9]. The oligomycins
contain analog isomers A through G that are highly selective for disrupting mitochondrial
metabolism [3, 4, 8, 10]. Although biological activities of oligomycins on fungi and the oomy-
cete *P. viticola* have been reported, very little is known about the effect of these natural prod-
ucts on the notorious wheat blast fungus *Magnaporthe oryzae Triticum* (MoT). The
bioactivities of oligomycins against different classes of fungal species indicates that their targets
may involve a variety of cellular processes, such as inhibition of mycelial growth of *Cladospor-
ium cucumerinum*, *Magnaporthe grisea*, *Colletotrichum lagenarium*, *Botrytis cinerea*, *Cylindro-
carpon destructans*, *Fusarium culmorum*, *Erysiphe graminis* and *Phytophthora capsici* [3, 11],
lysis and motility inhibition of *P. viticola*, and *Aphanomyces cochlioides* zoospores [4].

The wheat blast fungus MoT is one of the most destructive pathogens of wheat [12–15].
The three-celled, hyaline and pyriform fungal conidium attaches to the host surface by
secreted adhesive [14, 16, 17]. The attached conidium germinates to form a hyphal germ tube,
an appressorium and a penetration peg to penetrate the epidermis of the host and complete
the infection process [16, 18]. The invasion of plant tissue is achieved by penetrating the epi-
dermal cells and invaginating the host plasma membrane [16–18]. The fungus can attack
wheat plants at any stage of development and infects leaves, nodes, stems, and spikelets [15, 17,
19]. Mycelium can survive in the embryo, endosperm, and kernal tissues of wheat seed. Wheat
blast mainly affects wheat heads; it bleaches the infected heads, resulting in deformed seed or
no seed production [14]. The badly affected wheat heads can die, leading to a drastic reduction
in grain yield. Bleaching of the spikelets or the entire head at an early stage is the most com-
mon recognizable symptom of the disease [12, 14, 15]. Infected seeds and airborne conidia
usually disseminate the fungus which may survive in infected seeds and crop residues [20].
Pyriform conidia developed from conidiophores and conidia germination with appressorial
development at the germ tube tips are essential steps of the disease cycle of MoT [16]. Disrup-
tion of any of these asexual life stages reduces the chance of pathogenesis and development of
an epidemic [21]. Finding natural bioactive compounds capable of inhibiting any of these
asexual life stages is considered the first step in the development of a new fungicide for control-
ling MoT.

Wheat blast was first found in Brazil in 1985, and subsequently spread to neighboring
Bolivia and Paraguay [12, 19]. The cultivation of wheat (*Triticum aestivum*) has increased in
Bangladesh in recent years making it the 2nd largest food source after rice. A sudden outbreak
of wheat blast occurred in Bangladesh in 2016, which was the first incident of wheat blast out-
side of South America [14, 22]. About 15,000 hectares of wheat were destroyed, resulting in
about 15% crop losses in Bangladesh [14]. The outbreak concerned crop scientists as it has the
potential to extend further to major wheat-producing regions in neighboring South Asian
countries and Africa due to similar climatic conditions [23]. Plant pathologists have cautioned
that this disease is expected to disperse to India, Pakistan and China, that rank 2nd, 8th and 1st,
respectively, in the world for wheat production, [15, 24].

Current wheat blast disease management methods include the utilization of synthetic fungicides. Natural products generally impart less harmful effects on the environment and health of living species including humans when compare to their synthetic contemporaries [25, 26]. Indiscriminate application of synthetic commercial fungicides for plant protection may also result in development of resistance in fungal population to fungicides [13, 17]. In Brazil and other South American countries, some MoT strains have developed resistance to strobilurin (QoI) and triazole fungicides [27, 28]. Nowadays, natural products that are environment-friendly with minimum toxicity to living organisms are gaining popularity as important ecologically suited alternative fungicides for protecting plants. Therefore, search for novel bioactive natural products against MoT is an urgently needed scientific endeavor.

The biological approach for plant disease management offers a better alternative to the control of wheat blast. There is very little information available on the antagonistic effects of *Streptomyces* spp., and/or secondary metabolites derived from them to control wheat blast. We screened 150 natural compounds belonging to the classes alkaloids, terpenoids, macrolides, macrotetrolides, tepenoids, and phenolics isolated from different plants and microorganisms for antagonistic activity against MoT in our laboratory [21]. Among them, the two most potential macrolides, oligomycin B and oligomycin F, previously extracted from the marine *Streptomyces* spp. [3–4] were selected for this study. The specific objectives of this study were to: (i) test the effect of oligomycin B and F on the inhibition of mycelial growth of MoT; (ii) examine their effect on conidiogenesis, conidial germination and subsequent morphological development; (iii) evaluate the suppression of wheat blast disease using detached wheat leaves; and iv) compare the efficacy of these two oligomycins with a local standard fungicide product.

## Materials and methods

### Chemicals

Oligomycin B and oligomycin F (Fig 1) were isolated from the marine bacteria, *Streptomyces* sp. strains B8496, B8739 and A171 [3–4]. These pure compounds were generously provided by Dr. Hartmut Laatsch of Georg-August University Goettingen, Germany. The fungicide Nativo® WG 75 (a combination of tebuconazole, 50% and trifloxystrobin, 25%) was purchased from Bayer Crop Science Ltd. Dhaka, Bangladesh.

### Fungal strain, growth media and plant materials

The strain BTJP 4 (5) of MoT was isolated from blast infected spikelets of wheat cv. BARI Gom-24 (Prodip) in Jhenaidah, Bangladesh in 2016. For this research a pure culture from a

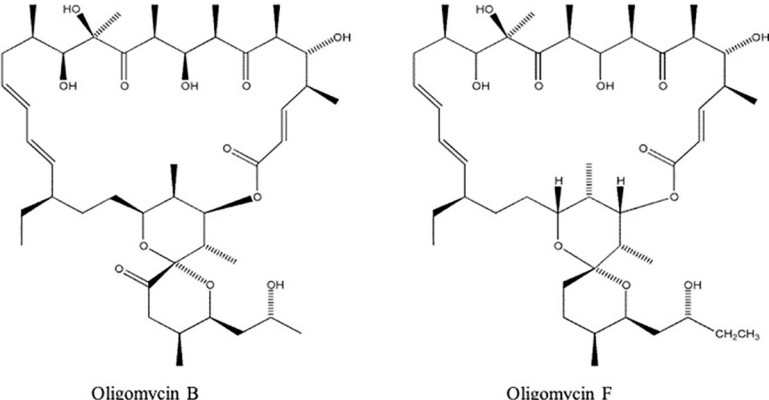

**Fig 1. Structures of oligomycin B and F.**

single spore was preserved at 4°C on dry filter paper [14]. The isolate (BTJP 4) was re-cultured on Potato Dextrose Agar 42 g/L (PDA) at 25°C for 7 days. Ten-day-old fungal cultures grown on PDA were washed in an aseptic environment in a laminar flow hood with 500 ml of deionized distilled water to remove aerial mycelia; then kept at ambient room temperature (25–30°C) for 2–3 days to induce abundant conidia production [14, 29]. Conidia were scraped from the plates with a glass slide after adding 15 ml water into each plate. The conidial and mycelial suspension was filtered through two layers of cheese cloth and adjusted to a concentration of $1 \times 10^5$ conidia/ml. Conidial germination was visualized and counted under a compound microscope. Wheat blast susceptible, five-leaf stage seedlings of cultivar BARI Gom-24 (Prodip) was used for the leaf bioassay [30].

## Inhibition of mycelial growth and morphological effects on hyphae

A modified disk diffusion technique [31] was used to determine hyphal growth suppression of MoT isolate BTJP by the oligomycins and the commercial fungicide, Nativo ® WG. A series of concentrations ranging from 0.005 to 2 μg / disk of the oligomycins and the fungicide Nativo ® WG75 were prepared by dissolving required amounts in ethyl acetate and water, respectively. Filter-paper disks (Sigma-Aldrich Co., St. Louis, MO, USA) measuring nine-millimeter diameter were soaked with the test compounds. The treated disks were placed 2 cm from one side of 9 cm dia Petri dishes containing 20 ml PDA. Five-millimeter diameter mycelial plugs from actively growing seven-day-old PDA cultures of MoT were placed on the opposite side of filter paper disk containing test compounds. Petri dishes inoculated with fungal mycelial plugs against fungicide Nativo ® WG75 were used as an industry standard. Filter paper disks treated with ethyl acetate followed by evaporation of ethyl acetate in room temperate served as a negative control. Inhibition of fungal growth was apparent within 10 days of incubation. Plates were incubated at 25°C until the fungal colony fully covered the agar surface of the control plates. There were five replications for each concentration and the experiment was repeated five times. The fungal colony's radial growth was measured in centimeters with a ruler along with two perpendicular lines drawn on each plate's lower side. Data were recorded by measuring the inhibition zone created by test compounds and corresponding mycelial growth. Radial growth inhibition percentage (RGIP) (± standard error) [32] was calculated from mean values as:

$$\text{RGIP (\%)} = \frac{\text{Radial growth in control plate} - \text{Radial growth in treated plate}}{\text{Radial growth of control}} \times 100$$

Hyphal morphology at the leading edge of the colonies facing the treated and control disks were observed with a Zeiss Primo Star microscope at 40X and 100X (100x was an oil emersion lens). Images of the disk diffusion experiment were captured with a Canon DOS 700D digital camera. Images of the hyphae were captured with a Zeiss Axiocam ERc 5s through the microscope.

## Inhibition of conidiogenesis

Stock solutions of each of the oligomycins were prepared in 10 μl of dimethyl sulfoxide (DMSO). Stock solution was then diluted with distilled water to obtain 5, 10 and 100 μg / ml concentrations. The final concentration of DMSO was never higher than 1% (v / v) in the final solution, which does not affect the hyphal growth or sporulation of MoT. Preparation of 5 ml fungicidal suspension of Nativo®WG75 at 5, 10 and 100 μg/ml concentrations was carried out by mixing the required amount of product in distilled water for using it as a positive control. Mycelium of a 10-day-old Petri dish culture of MoT was washed to reduce nutrients and

induce conidiogenesis [14, 29]. Ten mm MoT mycelial agar blocks were treated with 50 μl of each compound and Nativo®WG75 at 5, 10 and 100 μg/ml and put into Nunc multi well plates. The same amount of sterile water was applied on the MoT mycelial block with 1% DMSO serving as a negative control. Treated mycelial agar blocks MoT were incubated at 28˚C with >90% RH and 14 h light followed by 10 h of darkness. After 24 hours, conidiogenesis was observed with a Zeiss Primo Star microscope at 40x magnification and images captured with a Zeiss Axiocam ERc 5s. There were five replications per treatment and the experiment was conducted five times.

## Inhibition of conidial germination and morphological changes of germinated conidia

A stock solution of each oligomycin was prepared by dissolving 0.1 μg of the compound in 10 μl dimethyl sulfoxide (DMSO) followed by diluting the concentration. of each compound to 0.1μg/ml by adding distilled water. Nativo®WG75 solution was prepared with distilled water at 0.1 μg/ml to use as a positive control. Conidial germination assays were carried out following the protocol described by Islam and von Tiedemann [33]. For each treatment, a 100 μl solution of 0.1 μg/ml was added directly to 100 μl of $1 \times 10^5$ conidia/ml of MoT to make a final volume of 200 μl containing 0.05 μg/ml test compound into a well of a 96-multiwell plate. The solution was mixed immediately with a glass rod and incubated at 25˚C. Sterilized water with 1% DMSO served as a control. The multiwell plate was incubated in a moisture chamber at 25 $^0$C for 6 h, 12 h and 24 h in the dark. A total of 100 conidia from each of five replicates were examined under a Zeiss Primo Star at 100x magnification. Percent germination of conidia and developmental differences of the germ tubes and appressoria were evaluated and the images were captured with a Zeiss Axiocam ERc 5s. Each treatment and time course was replicated five times and the experiments repeated five times. The percent conidial germination (± standard error) was calculated from mean values as: CG % = (C–T)/C × 100; Where, CG = conidial germination, C = percentage of germinated conidia in control, and T = percentage of germinated conidia in treated samples.

## Development of wheat blast on detached wheat leaves

Stock solutions of oligomycins B and F were prepared using dimethyl sulfoxide (DMSO). Then preparation of 5, 10 and 100 μg/ml concentrations of each compound was carried out in distilled water where the final concentration of DMSO never exceeded 1%. Nativo®WG75 concentrations were 5, 10 and 100 μg/ml. Sterilized water with 1% DMSO served as a negative control. Wheat leaves were separated from five-leaf stage seedlings and placed in plates lined with moist paper towels. Three 20-μl droplets of the freshly prepared test compounds at concentrations mentioned above were placed on three different spots of each leaf, and left for 15 minutes to dry. Each spot was then inoculated with 1 μl conidial suspension containing $1 \times 10^5$ MoT conidia/ml followed by incubating dishes at 28˚C under 100% relative humidity in dark for first 30 h, then 2 days in continuous light. The test was performed five times independently with 5 replicate samples. The resulting length of wheat blast lesions MoT were measured from 3 leaves per experiment for each treatment and each concentration of compounds.

## Statistical analysis, experimental design/replications

Experiments were performed using a completely randomized design (CRD) to determine biological activities of the pure oligomycin compounds compared to a standard fungicide. Data were analyzed by one-way ANOVA, and mean values were separated by the posthoc statistic of Tukey's HSD (honest significant difference). All statistical analyses were carried out with

SPSS (IBM SPSS statistics 16, Georgia, USA) and Microsoft Office Excel 2010 program package. Mean value ± standard error of 5 replications were used in Tables and Figures.

## Results

### Inhibition of mycelial growth and morphological effects on hyphae

Both oligomycins B and F (Fig 1) tested in this study and originally extracted from a *Streptomyces* species showed significant inhibition of MoT hyphal growth on PDA (Fig 2). Between these two compounds, oligomycin F depicted stronger inhibition of hyphal growth of MoT. Mycelial growth inhibition by oligomycins B and F was 57.1 ± 1.3% and 73.9 ± 2.5%, respectively when both compounds were used at 2 μg/disk (Fig 3) The commercial fungicide Nativo® WG 75 had a higher inhibition capacity (81.9 ± 0.9% at 2 μg/disk) than both oligomycin F and B.

Both oligomycins inhibited MoT mycelial growth in a dose dependent manner. Suppressive effects of oligomycin increased with increasing concentrations from 0.005 to 2 μg/disk reaching 74% for oligomycin F (Fig 3). Suppression by oligomycin F was slightly lower than suppression by Nativo® WG 75 but higher than oligomycin B. Neither of the oligomycins showed activity against MoT at concentration lower than 0.005 μg. Oligomycin F showed extensive inhibition of hyphal growth at 2 μg/disk (73.9 ± 2.5%) followed by 1.5 μg/disk (67.6 ± 0.9%) and 1 μg/disk (60.9 ± 2.5%) showing a positive correlation of suppression with an increase in concentration. The percent suppression by oligomycin B was 57.1 ± 1.3%, 54.3 ± 1.3% and 53.3 ± 1.5%, at 2, 1.5 and 1 μg/disk, respectively. The minimum inhibitory concentrations of oligomycin F and oligomycin B were 0.005 and 0.05 μg/disk, respectively. At the minimum inhibitory concentrations, hyphal growth inhibition was 11.4 ± 2.3% and 8.63 ± 1.3%, respectively for oligomycin F and B. The minimal inhibitory concentration of Nativo® WG 75 was 0.05 μg/disk, similar to oligomycin B although fungicide at higher concentration starting from 0.25 μg/disk superseded inhibition percentage at equivalent concentrations of oligomycins. It is interesting to note that at concentrations below 0.25 μg/disk, the inhibition of mycelial growth by oligomycin F was higher than that of the fungicide Nativo® WG 75 and this macrolide displayed inhibitory activity against MoT at about 10-fold lower concentration.

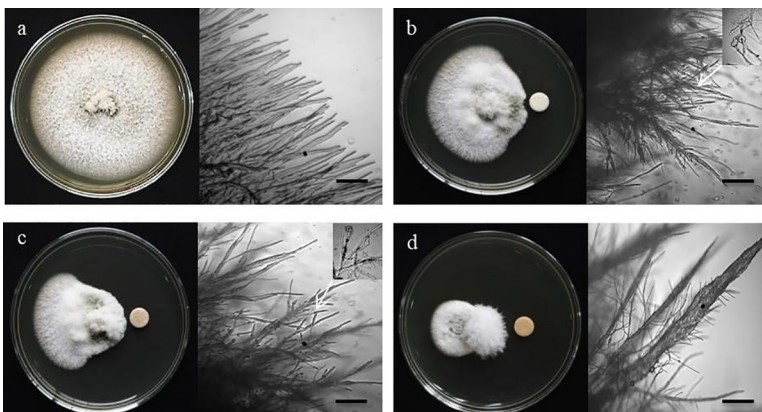

**Fig 2.** Macroscopic and microscopic images of *in vitro* antifungal activity of oligomycin B, oligomycin F and the commercial fungicide nativo® WG75 against *Magnaporthe oryzae Triticum* at 2 μg/disk; (a) Control, (b) Oligomycin B, (c) Oligomycin F, (d) Nativo® WG75. Bar = 10 μm, 50 μm.

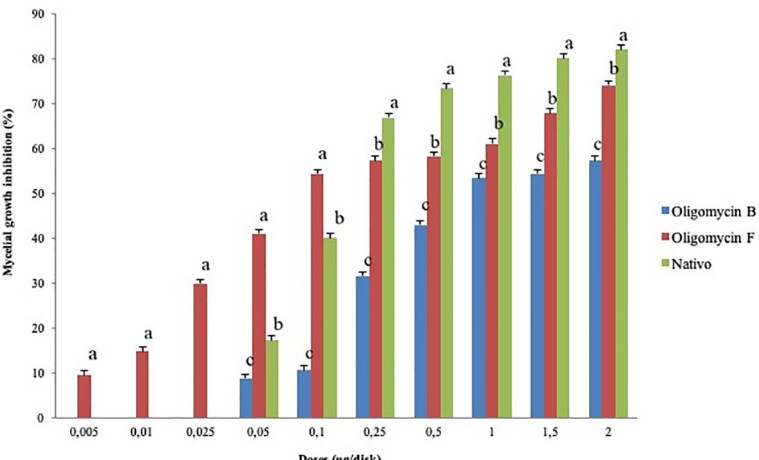

**Fig 3. Inhibitory effects of oligomycin B, oligomycin F and the commercial fungicide Nativo® WG75 on hyphal growth of *Magnaporthe oryzae Triticum* in potato dextrose agar.** The data are the mean ± standard errors of five replicates for each concentration of the compound tested at a 5% level based on the Tukey HSD (Honest Significance Difference) post-hoc statistic.

Microscopic observations showed that untreated MoT hyphae had polar, tubular growth with smooth, branched, hyaline, plump, septate, and intact hyphae (Fig 2A). Hyphae treated with oligomycin B and F showed irregular growth and an increase in branch frequency per unit length of the hyphae. The hyphal cell walls were not smooth but showed ridges giving a corrugated appearance and irregular swelling of cells (Fig 2B and 2C). Nativo ® WG75 showed a similar pattern of hyphal growth inhibition. Similar abnormality of MoT also occurred where the hyphae were close to the Nativo ® WG75 disk (Fig 2D). However, morphological changes of MoT by the two oligomycins were slightly different from those observed with the Nativo®WG75 suggesting a possible different mode of action.

## Inhibition of conidiogenesis

Conidial formation by MoT was remarkably decreased by the oligomycins and the fungicide at 5 and 10 μg/ml when compared to the control, and inhibition increased with an increase in concentration from 5, 10 and 100 μg/ml (Fig 4). For all three treatments, no conidia developed at 100 μg/ml. Microscopic examination revealed broken mycelial tips, and a complete lack of conidiophores for all three treatments at 100 μg/ml.

## Inhibition of conidial germination and morphological changes of germinated conidia

Oligomycin B, F and Nativo ® WG75 at 0.05 μg/ml were used to determine the inhibition of conidial germination of MoT in multi-well plates. After 6, 12 and 24 h of incubation, the percent of germinated conidia was recorded (Table 1). After 6 h, all three treatments significantly reduced germination of conidia compared to the control. Germination was 100% in water, and 50.3±0.7% in plates treated with Nativo ® WG75. With oligomycin B and F germination percentages of MoT conidia were 24 ± 0.9% and 53±0.4% at 0.05 μg/ml, respectively.

Conidial germination in water was 100% with normal germ tube development and mycelial growth at all incubation times (6 h, 12 h and 24 h) at 25˚C in dark (Table 1, Fig 5A). The two oligomycins had adverse effects not only on conidial germination but also post-germination developmental processes with abnormal transitions from one step to the next at 0.05 μg/ml.

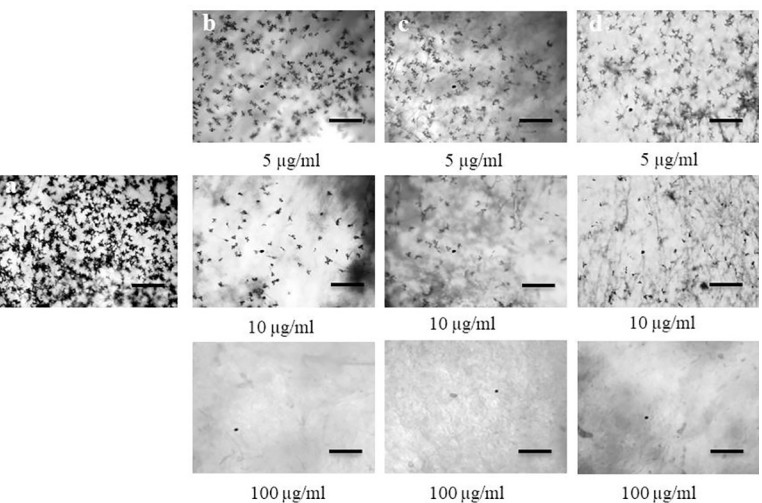

**Fig 4. Effects of oligomycin B, oligomycin F and the fungicide Nativo® WG75 on inhibition of conidiogenesis of** *Magnaporthe oryzae Triticum* **in 96-multiwell plates at 5 µg/ml, 10 µg/ml, 100 µg/ml.** (a) Control, (b) Oligomycin B, (c) Oligomycin F, (d) Nativo® WG75. Bar = 50 µm.

Out of 38% of the germinated spores in the oligomycin B treatment, 24 ± 0.9% had short germ tubes and 14.3± 0.7% of the germ tubes lysed within the first 6 h of incubation. After 12 h of incubation in the same treatment, out of 24% germinated spores, 10.6 ± 0.7% were normal, 8.4 ± 0.2% had shorter germ tubes than the control and 5 ± 0.4% had abnormally elongated germ tubes. After 24 h of incubation, no conidia germinated (Table 1, Fig 5B). In the presence

**Table 1. Effects of oligomycins and the fungicide Nativo® WG75 on germination of conidia and morphology of germ tubes and appressoria of** *Magnaporthe oryzae Triticum* **at 0.05 µg/ml** *in vitro*.

| Compound | Time (h) | Germination of conidia, and morphology of germ tubes and appressorial formation | |
|---|---|---|---|
| | | Germinated conidia (% ± SE[a]) | Morphological change/developmental transitions in the treated conidia |
| Water | 0 | 0 ± 0e | No germination |
| | 6 | 100 ± 0a | Germination with normal germ tube and normal appressoria |
| | 12 | 100 ± 0a | Normal mycelial growth |
| | 24 | 100 ± 0a | Normal mycelial growth |
| Oligomycin B | 0 | 0 ± 0e | No germination |
| | 6 | 38.3 ± 0.7d | 24 ± 0.9% Short germ tube and 14.3 ± 0.7% conidia lysed |
| | 12 | 24 ± 0.9d | 10.6 ± 0.7% Normal germ tube, 8.4 ± 0.2% short and 5 ± 0.4% Abnormally elongated germ tube |
| | 24 | 0 ± 0b | No appressoria, no mycelial growth |
| Oligomycin F | 0 | 0 ± 0e | No germination |
| | 6 | 59 ± 0.8b | 53 ± 0.4% Short germ tube and 6 ± 0.8% conidia lysed |
| | 12 | 53 ± 0.4b | 33.3 ± 0.5% Normal germ tube and 19.7 ± 0.2% abnormal branching at the tips |
| | 24 | 0 ± 0b | No appressoria, no mycelial growth |
| Nativo | 0 | 0 ± 0e | No germination |
| | 6 | 50.3 ± 0.7c | Germinated with a short germ tube |
| | 12 | 50.3 ± 0.7c | Normal germ tube |
| | 24 | 0 ± 0b | No appressoria; no mycelial growth |

[a]The data presented here are the mean value ± SE of three replicates in each compound. Means within the column followed by the same letter(s) are not significantly different from those assessed by Tukey's HSD (Honest Significance Difference) post-hoc (p ≤ 0.05). Conidia germination percent at different incubation times is not cumulative, rather at different time intervals.

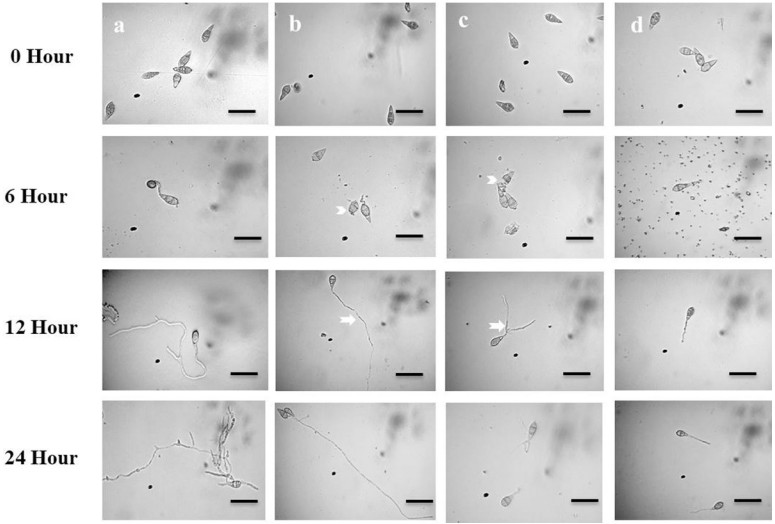

**Fig 5. Time-dependent alterations in *Magnaporthe oryzae Triticum* germination of conidia and subsequent morphological changes in the presence of oligomycin B, oligomycin F and the commercial fungicide Nativo® WG75.** Dose of oligomycins was 0.05 μg/ml. (a) Control, (b) Oligomycin B, (c) Oligomycin F, (d) Nativo® WG75. Branched germ tube (*arrow*); Elongated germ tube (*arrow*); Lysis of conidia (*arrow head*). Bar = 10 μm.

of oligomycin F, 53 ± 0.4% of conidia germinated with shorter germ tubes than the control and 6 ± 0.8% conidia lysed after 6 h. Similar developmental abnormalities were found among the germinated conidia after 12 h of incubation with 33.3 ± 0.5% normal and 19.7 ± 0.2% with abnormally branched germ tubes formation, while no germination after 24 h (Table 1, Fig 5C). In the presence of Nativo® WG75, 50.3 ± 0.7% conidia germinated with normal germ tubes after 6h and 12 h, but no appressoria developed. Nativo® WG75 also prevented spore germination after 24 h (Table 1, Fig 5D). It is interesting that the oligomycins produced abnormally long or short or branched germ tubes and lysing of conidia, while the fungicide did not result in these changes.

## Development of wheat blast on detached wheat leaves

Application of the two oligomycins at 5, 10 and 100 μg/ml remarkably inhibited symptoms of wheat blast in detached wheat leaves, inoculated with MoT. The average length of lesions in the wheat leaves treated with oligomycin B were 6.3 ± 0.3 mm and 1.8±0.3 mm at 5μg/ml and 10 μg/ml, respectively (Fig 6A and 6B). With oligomycin F and Nativo®WG75, blast lesion lengths were 4 ± 0.3 mm and 2 ± 0.3 mm at 5 μg/ml, respectively (Fig 6A and 6B). No blast symptoms were visible when leaves were treated with oligomycin F and the fungicide Nativo®WG75 at 10 μg/ml and 100 μg/ml (Fig 6A and 6B). No visible blast lesion occurred when treated with oligomycin B at 100 μg/ml. Leaves treated with water as a negative control developed typical blast lesions with an average length of 9.58 ± 0.2 mm (Fig 6A and 6B). These results show that the fungicide suppressed lesion development more than both oligomycins at 5 μg/ml, the fungicide showed more suppression than oligomycin F at 10 μg/ml, and no lesions developed with any of the three treatments at 100 μg/ml.

## Discussion

In this study, we found that two *Streptomyces* macrolides, oligomycin B and oligomycin F, demonstrated extensive antifungal activities against the devastating wheat blast pathogen of

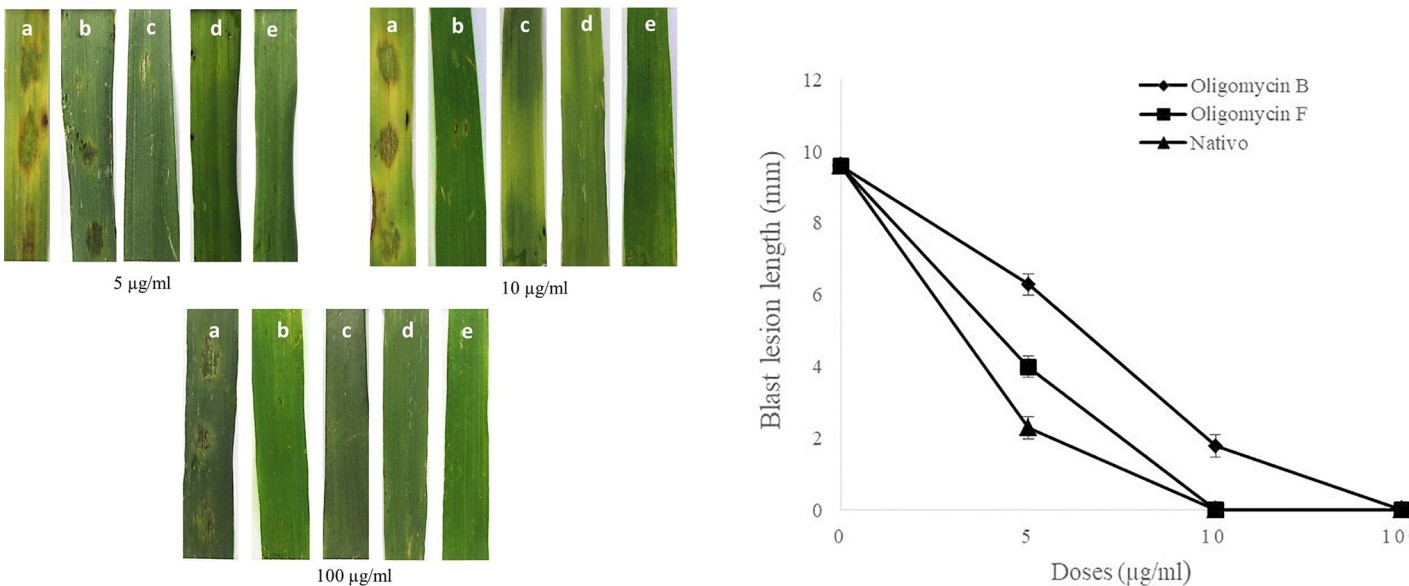

**Fig 6.** Suppression of wheat blast symptoms with oligomycins at 5 μg/ml, 10 μg/ml and 100 μg/ml on a representative detached wheat leaf of five replicates inoculated with *Magnaporthe oryzae Triticum* (A) blast lesions on treated and untreated wheat leaves (a) Water control+MoT, (b) Oligomycin B+MoT inoculation, (c) Oligomycin F+MoT inoculation, (d) Nativo® WG75+MoT inoculation, (e) Non-inoculated, non-treated leaf; (B) Blast lesion lengths on detached wheat leaves treated with oligomycin B oligomycin F and Nativo ® WG75 fungicide compared with water treatment control. The data are the averages ± standard errors of at least five replicates for each dose of the tested compounds at p ≤ 0.05. Bars represent ± standard error.

wheat. Of the two oligomycins tested, biological activity of oligomycin F was superior than oligomycin B and in some cases, superior to the commercial fungicide Nativo® WG75. These findings indicate that suppression of conidial germination, appressorial formation and mycelial growth by these macrolides are correlated with wheat blast disease suppression on inoculated leaves. Hyphal growth inhibition, conidial formation and germination of conidia of various fungi including the rice blast fungus (MoO) by a number of natural secondary metabolites from *in vitro* bioassays have been documented by many investigators [3, 11, 34–39]. To the best of our knowledge, this is the first report of suppression of the devastating wheat blast fungus by oligomycin B and F isolated from the *Streptomyces* spp., with the potential to be utilized for managing the disease *in vivo*.

Oligomycins are macrolide antibiotics that block the proton channel (F0 subunit) requisite for the oxidative phosphorylation of ADP to ATP due to inhibition of ATP synthase. [40]. While oligomycins have excellent biological properties, only a few studies have so far focused on the development of plant disease protection products from these macrolides. Interestingly, our result also revealed that the efficacy of oligomycin F in controlling wheat blast fungus was 10-fold stronger than the commercial fungicide Nativo ® WG75 in terms of mycelial growth inhibition.

One of the noteworthy findings from the present study is the induction of swelling on the MoT hyphae by these macrolides (Fig 2B and 2C), which is often considered as a reliable mode of inhibitory action of a compound against the normal growth and development of a fungal pathogen [38, 39]. We tested a range of concentrations of these compounds from 0.005 to 2 μg /disk, and found that swelling in hyphae increased with increasing concentrations of the oligomycins (data not shown). Swelling in various fungal hyphae has been reported earlier by polyoxin B [41], fengycin [37–39] and tensin [42]. Morphological changes like extensive branching and swelling of hyphae of an oomycete pathogen *Aphanomyces cochlioides* by phloroglucinols

extracted from *Pseudomonas fluorescence* or xanthobaccin A isolated from *Lysobacter* sp. SB-K88 have been documented [43–46]. According to a report by Kim et al. [11], oligomycin A from *Streptomyces libani*, significantly inhibited mycelial growth of *Magnaporthe grisea*, *Botrytis cinerea*, *Colletotrichum lagenarium*, *Cylindrocarpon destructans*, *Cladosporium cucumerinum* and *Phytophthora capsici*. So far, this is the first report of swollen-like structures development in hyphae by oligomycins toward the destructive phytopathogen, MoT. Further investigation is needed to understand the detailed modes of action of these macrolides towards suppression of the phytopathogen, MoT.

Conidia are the infecting propagule by which most pathogenic fungi invade plants and the process by which conidia are formed is known as conidiogenesis [35, 47]. The more conidia a fungal pathogen produces the more its potential to destroy a plant, which is very significant in case of an economically important cereal crop like wheat. Inhibition of conidiogenesis and conidial germination reduces the chance of secondary infection. Compounds that are found to inhibit these processes are great candidate for downstream application as plant protection products. A few interesting findings of this study showed that both oligomycin B and F not only strongly suppressed conidiogenesis (Fig 4), but also inhibited germination of conidia and further morphological advancement of the germ tube towards hyphal growth (Table 1, Fig 5). Results from the bioassay showed that wheat leaves prophylactically treated with these macrolides at 5, 10 and 100 μg/ml had restricted fungal growth and limited disease development. Other novel and interacting phenomena observed in this study included lysis of conidia, irregular branching of the tip of germ tube, and abnormally elongated hypha-like germ tubes (Fig 5B and 5C). Similar phenomenon was observed by Dame et al., [4] who reported that oligomycin A, B and F from a marine *Streptomyces* could induce lysis of zoospores of the plant pathogen *Plasmopara viticola*, which causes grapevine downy mildew. Homma et al. [48] reported that lecithin induced abnormal branching at the conidial germ tube tips and inhibited appressoria formation of rice blast fungus. Islam and Fukushi [46] reported that cystospores of *A. cochlioides* produced in the presence of diaacetylphloroglucinol (DAPG) subsequently germinated with hyperbranched germ tubes. Modes of action and mechanism of inhibition of conidiogenesis, germination and formation of appressoria of the MoT conidia by oligomycins has not been previously reported.

Oligomycins are macrolide antibiotics that impede ATP production by influencing oxidative phosphorylation in mitochondria [49]. The oligomycin comprises a 26-membered α, β-unsaturated lactone with a conjugated diene fused to a bicyclic spiroketal ring system. Their mode of action includes the decoupling of mitochondrial ATPase F0 and F1 factors responsible for promoting the transfer of proton via the inner mitochondrial membrane [50]. The enzymatic complex F0F1 ATP synthase may be considered as a target for antifungal and anti-tumor or anti-infection therapy [51]. Oligomycins display a number of important biological activities including mitochondrial ATPase inhibition, strong antifungal, anti-actinobacterial and anti-tumor effects that have been reported [9, 1, 51]. These natural products are among the strongest selective agents in the cell line; they interrupt P-glycoprotein activity and induce apoptosis in doxorubicin-resistant HepG2 cells [52]. Oligomycins have a variety of isomers called oligomycin A through G. These are particularly relevant to the disruption of mitochondrial metabolism [10]. Moreover, elucidation of structure has provided new horizons for developing new ATP synthase-directed agents with possible therapeutic effects [53]. The first reports of chemical modification of oligomycin A have already been documented by Lysenkova et al. [54]. New compounds also showed efficacy against *Candida albicans*, *Aspergillus niger* and *Cryptococcus humicolus*, with other biological properties similar to those of oligomycin A, but with less cytotoxic effects. During germination, conidia might need a constant energy (ATP) supply from the internal energy reserve of the cells [55]. Therefore, a plausible

explanation for the suppression of hyphal growth and conidia germination of MoT demonstrated in this study is likely to be associated with ATP synthesis inhibition in mitochondria due to the effects of oligomycins. More studies are required to determine the precise structure-activity relationships of these oligomycins, which may make it possible to synthesize a more active oligomycin as an effective agrochemical against MoT.

A hallmark finding of this study is that application of both macrolides significantly inhibited blast disease development in detached leaves of wheat (Fig 6). In this study, wheat leaves treated with oligomycin B and F had shorter lesion lengths than the untreated control (Fig 6). Many of the lesions in treated leaves were small brown in color with pinhead-sized specks (scale 1) to small, roundish to slightly elongated infecting <10% of wheat leaf area (scale 5). In contrast, water treated control leaves had typical blast lesions infecting more than 75% wheat leaf area (scale 9) corresponding to scale 9 of blast disease assessment scale provided by the IRRI Standard Evaluation System [56]. However, no blast lesions were visible on the leaves treated with the oligomycin compounds and Nativo ® WG75 at the highest concentration (Fig 6). Nativo ® WG75 is a systemic wide-spectrum commercial fungicide that we used as a positive control. Interestingly, the antifungal effect of oligomycins on the inhibition of MoT fungus was found equivalent or stronger than that of the fungicide. Tebuconazole and trifloxystrobin are two main active ingredients of Nativo ® WG75. Tebuconazole is known as a demethylase inhibitor (DMI), which is a systemic triazole fungicide. Demethylase inhibitors inhibit ergosterol biosynthesis, which is a major component of the plasma membrane of certain fungi essential for growth and further development of the fungus [57]. Trifloxystrobin is a strobilurin fungicide that interferes with the respiration of plant pathogenic fungi by preventing energy production in mitochondria, thereby inhibiting germination of fungal conidia [58]. The mechanisms of disease suppression by the oligomycins are likely different compared with the mechanisms of Nativo ® WG75 although similar disease suppression effect has been obtained. Further study is needed to elucidate the underlying mechanism of wheat blast disease suppression by the oligomycin B and F. Furthermore, a field trial of the oligomycins in controlling wheat head infection is needed before considering them as effective fungicides against the wheat blast.

Despite their significant potency as antifungal compounds, little information is available on the effectiveness of oligomycins as agricultural fungicides. The shorter residual effect of oligomycin can be important in the reduction of deleterious effects on humans and the environment, considering that sufficient efficacy in the management of plant diseases is sustained [11]. Oligomycin A was found the most active anti-filamentous fungal analogue among antibiotics in the oligomycin family [1, 59, 60]. As oligomycin F is the immunosuppressive homolog of oligomycin A [3], and oligomycin B is a stable natural product [61], these macrolides have the potential to be the leading compounds for the production of agrochemicals against the cereal killer MoT.

Frequent application of commercial fungicides with site-specific modes of action, like strobilurins (QoI, quinone outside inhibitors) and triazoles, has resulted in the widespread occurrence of resistant mutant species in MoT [27, 28]. Resistance development in fungal population against fungicides prompted search for new, effective antifungal agents with alternate mode of action to protect wheat plants against this phytopathogenic fungus. Results from this study pertaining to the inhibitory ability of these macrolides should motivate agrochemical companies to consider these as candidates for commercial products with novel modes of action against the wheat blast fungus.

## Conclusion

Our findings show that oligomycin B and F from *Streptomyces* spp., suppress hyphal growth and asexual development of MoT, and inhibited wheat blast development on detached leaves of wheat. Field assessment of these macrolides is required to evaluate these metabolites as effective fungicides against wheat blast. Further research is also required to understand the mode of action and the structure-activity relations among oligomycins A-G against the devastating wheat killer, *M. oryzae Triticum*.

## Acknowledgments

The authors are also thankful to Dr. Hartmut Laatsch of Georg-August University Goettingen, Germany, for kindly providing the oligomycins for this research. Our sincere thanks are due to Dr. Mahfuzur Rahman and Tahsin Islam Sakif of West Virginia University, USA for linguistic editing of this manuscript.

## Author Contributions

**Conceptualization:** Tofazzal Islam.

**Data curation:** Moutoshi Chakraborty, Nur Uddin Mahmud, Tofazzal Islam.

**Formal analysis:** Moutoshi Chakraborty, Nur Uddin Mahmud, Tofazzal Islam.

**Funding acquisition:** Tofazzal Islam.

**Investigation:** Moutoshi Chakraborty, Nur Uddin Mahmud, Abu Naim Md. Muzahid, S. M. Fajle Rabby.

**Project administration:** Tofazzal Islam.

**Resources:** Tofazzal Islam.

**Software:** Moutoshi Chakraborty, Nur Uddin Mahmud.

**Supervision:** Tofazzal Islam.

**Visualization:** Nur Uddin Mahmud, Abu Naim Md. Muzahid, S. M. Fajle Rabby.

**Writing – original draft:** Moutoshi Chakraborty, Nur Uddin Mahmud, Tofazzal Islam.

**Writing – review & editing:** Tofazzal Islam.

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
