## [Decision Letter · Decision Letter 0]

24 Jun 2020

PONE-D-20-13723

Oligomycins Inhibit Magnaporthe oryzae Triticum and suppress wheat blast disease

PLOS ONE

Dear Dr. Islam,

Thank you for submitting your manuscript to PLOS ONE. After careful consideration, we feel that it has merit but does not fully meet PLOS ONE’s publication criteria as it currently stands. Therefore, we invite you to submit a revised version of the manuscript that addresses the points raised during the review process.

This was a timely and potentially important study focused on characterizing oligomycins with activity against the devastating wheat blast pathogen MoT. However, concerns were raised. Reviewer 1 noted that in order to determine the exact MIC, the oligomycins should be included in the liquid or solid media. Reviewer 2 made many suggestions for improving the manuscript, which were included in two separately uploaded text files. Please address all the Reviewer's comments and suggestions.

We look forward to receiving your revised manuscript.

Kind regards,

Richard A Wilson

Academic Editor

PLOS ONE

Journal Requirements:

2. Thank you for stating the following in your Competing Interests section: 'No competing interests'

a. Please complete your Competing Interests statement to state any Competing Interests.

If you have no competing interests, please state "The authors have declared that no competing interests exist.", as detailed online in our guide for authors at http://journals.plos.org/plosone/s/submit-now

Reviewers' comments:

Reviewer's Responses to Questions

**Comments to the Author**

1. Is the manuscript technically sound, and do the data support the conclusions?

Reviewer #1: Yes

Reviewer #2: Partly

2. Has the statistical analysis been performed appropriately and rigorously? 

Reviewer #1: Yes

Reviewer #2: Yes

3. Have the authors made all data underlying the findings in their manuscript fully available?

Reviewer #1: Yes

Reviewer #2: No

4. Is the manuscript presented in an intelligible fashion and written in standard English?

Reviewer #1: Yes

Reviewer #2: No

5. Review Comments to the Author

Reviewer #1: Chakraborty et al. showed the effects of two oligomycins on disease cycle of MoT. Although the antifungal effects of these molecules on the growth of M. grisea was shown before (ref 11), this manuscript includes their effects on disease phenotype including hyphae growth, conidiation, conidia mophology, and pathogenicity. In addition, there is urgent need to develop effective eradication methods because MoT is newly emerging on South Asia. Thus this manuscript is timely proper for publication on PLoS ONE. However, some points should be clarified before publication.

One major point is about MIC. The present method can show relative antifungal activiies only. But it is not proper for determination of MIC because only diffused oligomycins have effects on the edge of mycelium. We do not know exact amount of the effective molecules. For the exact MIC, the molecules should be included in the liquid or solid media. Or the authors can give up mentioning MICs in the manuscript.

Figure 5 might be conidia but more magnified and clear images should be given. And the concentration on the figure is not matched to description(296-298) in the manuscript.

The quality of Fig 6 and 7 is not good.

Figure 7B need to be drawn again scientifically.

Reviewer #2: Data support conclusions; they mostly do, however the "discussion" is very long and there are some statements that are not borne up by the data in the results. Is data fully available; mostly, weak points have to do with no data for swollen hyphae and whether shorter or longer germs tubes should be quantified by measurements. Standard English needed a lot of work. I am afraid some of my editing may have changed the author's meaning because I was at times unsure what was being conveyed.

6. PLOS authors have the option to publish the peer review history of their article (what does this mean?). If published, this will include your full peer review and any attached files.

Reviewer #1: No

Reviewer #2: No

---

## [Author Response · Author response to Decision Letter 0]

9 Jul 2020

Responses to the Reviewers Comments/Suggestions

1. Is the manuscript technically sound, and do the data support the conclusions?

Reviewer #1: Yes

Our response:

Thank you for the encouraging comment.

Reviewer #2: Partly

Our response:

Thank you for the comment.

2. Has the statistical analysis been performed appropriately and rigorously?

Reviewer #1: Yes

Our response:

Thank you for the encouraging comment.

Reviewer #2: Yes

Our response:

Thank you for the encouraging comment.

3. Have the authors made all data underlying the findings in their manuscript fully available?

Reviewer #1: Yes

Our response:

Thank you.

Reviewer #2: No

Our response:

Thank you for the comment. We have provided all the data underlying the findings described in our manuscript in Tables and Fugures.

4. Is the manuscript presented in an intelligible fashion and written in standard English?

Reviewer #1: Yes

Our response:

Thank you for the encouraging comment.

Reviewer #2: No

Our response:

Thank you for the comment. We have substantially revised the manuscript responding to the reviewer’s suggestions to make the manuscript clearer, accurate, and unambiguous. 

5. Review Comments to the Author

Reviewer #1: Chakraborty et al. showed the effects of two oligomycins on disease cycle of MoT. Although the antifungal effects of these molecules on the growth of M. grisea was shown before (ref 11), this manuscript includes their effects on disease phenotype including hyphae growth, conidiation, conidia mophology, and pathogenicity. In addition, there is urgent need to develop effective eradication methods because MoT is newly emerging on South Asia. Thus this manuscript is timely proper for publication on PLoS ONE. However, some points should be 

clarified before publication.

Our response:

Many thanks for these encouraging comments.

One major point is about MIC. The present method can show relative antifungal activities only. But it is not proper for determination of MIC because only diffused oligomycins have effects on the edge of mycelium. We do not know exact amount of the effective molecules. For the exact MIC, the molecules should be included in the liquid or solid media. Or the authors can give up mentioning MICs in the manuscript.

Our response:

Thank you for this valuable comment. We have deleted the term MIC and also Figure 4 from the manuscript.

Figure 5 might be conidia but more magnified and clear images should be given. And the concentration on the figure is not matched to description (296-298) in the manuscript.

Our response:

Thank you for the comment. We have provided representative images of the conidia taken after 24 hrs of incubation by Zeiss Primo Star microscope at 40x magnification. The images demonstrated the gradual decrease of sporulation with the increasing concentrations of compounds. 

Thank you for pointing out this erroneous placement of the images in the figure 5. We have revised Figure 5 for matching with the description (296-298) in the manuscript.

The quality of Fig 6 and 7 is not good.

Our response:

Thank you for this comment. We have provided representative images in Figure 6 and 7. We have replaced some images in Figure 6 for a better demonstration.

Figure 7B need to be drawn again scientifically.

Our response:

Thank you for this comment. We have drawn it scientifically.

Reviewer # 2: Data support conclusions; they mostly do, however the "discussion" is very long and there are some statements that are not borne up by the data in the results. 

Our response:

Thank you for this valuable comment. We have deleted redundant sentences from the discussion part to make it more precise.

Is data fully available; mostly, weak points have to do with no data for swollen hyphae and whether shorter or longer germs tubes should be quantified by measurements. 

Our response:

Thank you for this comment. We have given the picturesque data of swollen hyphal cells in Figure 2 captured by a camera (ZEISS Axiocam ERc 5s) attached to the microscope at 100x magnification according to the reviewer’s comment, and also described the data in the result portion of the manuscript. We have quantified shorter or longer germs tubes on the basis of observations.

Standard English needed a lot of work. I am afraid some of my editing may have changed the author's meaning because I was at times unsure what was being conveyed.

Our response:

Thank you for the comment. We sincerely appreciate the reviewer for the helpful tips and huge linguistic editing for the improvement of the manuscript. We have revised it substantially following your suggestions. We hope that you would have a pleasant experience in reading it now.

Additional editorial Comments/Suggestions

Line:

1: Title; the term pathotype has been used by other researchers for this taxon but it is not officially accepted; currently there is no formal acceptance of the taxon Magnaporthe oryzae triticum but this is evolving nomenclature. In any case I would not capitalize triticum. When taxa include the genera of plants in the binomial (or trinomial), they are not capitalized.

Our response:

With due respect with the reviewer comment, we would like to mention that a numbers of previous studies were based on Magnaporthe oryzae which published in reputed peer-reviewed journals. The authors of these studies used the pathotype name of M. oryzae with the capitalized Triticum. Here are some references for your kind consideration.

Pieck, M. L., Ruck, A., Farman, M. L., Peterson, G. L., Stack, J. P., Valent, B., & Pedley, K. F. (2017). Genomics-Based Marker Discovery and Diagnostic Assay Development for Wheat Blast. Plant Disease, 101(1), 103–109. doi:10.1094/pdis-04-16-0500-re

Gladieux, P., Condon, B., Ravel, S., Soanes, D., Maciel, J. L. N., Nhani, A., … Fournier, E. (2018). Gene Flow between Divergent Cereal- and Grass-Specific Lineages of the Rice Blast Fungus Magnaporthe oryzae . mBio, 9(1). doi:10.1128/mbio.01219-17 

Yasuhara-Bell, J., Pedley, K. F., Farman, M., Valent, B., & Stack, J. P. (2018). Specific detection of the wheat blast pathogen (Magnaporthe oryzae Triticum) by loop-mediated isothermal amplification. Plant Disease. doi:10.1094/pdis-03-18-0512-re 

Gupta, D. R., Avila, C. R., Win, J., Soanes, D. M., Ryder, L. S., Croll, D., … Islam, M. T. (2018). Cautionary Notes on Use of the MoT3 Diagnostic Assay for Magnaporthe oryzae Wheat and Rice Blast Isolates. Phytopathology. doi:10.1094/phyto-06-18-0199-le 

Tosa, Y., Tamba, H., Tanaka, K., and Mayama, S. 2006. Genetic analysis of host species specificity of Magnaporthe oryzae isolates from rice and wheat. Phytopathology 96:480-484.

14: “…and the oomycete Plasmopara viticola.” The reference that this comes from only indicates activity against this particular oomycete, and you cannot assume all “peronosporomyctes” are equally affected. The author of the cited paper took excessive liberties in using the term peronosporomycetes.

Our response:

Thank you for this comment. We have revised it as per your suggestion.

21: omit “disease” here and in other places where it follows “wheat blast”. Wheat blast is a disease so we don’t have to indicate that it is a disease.

Our response:

Thank you for this comment. We have revised it as per your suggestion.

22: while the term “mycelia” continues to crop up in recent manuscripts, I am not in favor of its use. Mycelium is plural, so why pluralize it again? “Mycelia” may be appropriate when you are speaking of mycelium from several different isolates or a collection, like “mycelia of Basidiomycetes”. You occasionally use the term “mycelium”, be consistent and use mycelium throughout.

Our response:

Thank you for this critical comment. We have revised it as per your suggestion.

30: “process of conidial formation” I don’t know what this means; state specifically what process you are talking about.

Our response:

Thank you for this comment. Here we used the term “process of conidia formation” in the place of “conidiogenesis”. Conidiogenesis is the process by which conidia formation occurred in the mycelium under a certain environmental conditions. We have revised it.

95: “Most of the synthetic fungicides are harmful…” This is an overstatement and certainly references cited (25-26) would not support the statement since that was not the nature of their research. These references may make that claim but repeating the claim would not be acceptable since these research papers were not about demonstrating fungicides are harmful to humans. 

Our response:

Thank you for the comment. We have revised the statement with appropriate references.

100: “Nowadays, natural products…” This is a spurious argument considering the fungicides that are used for wheat blast have an LD 50 that are well more than twice as high as the LD 50’s of oligomycins. It’s not good enough to simply state that natural products are safer than synthetic ones. 

Our response:

Thank you for the comment. We have revised the sentence.

It is interesting to note that strobilurins are analogs of natural products.

Our response:

Thank you for the comment. Strobilurins are a group of natural products and their synthetic analogs which are are used in agriculture as fungicides (https://en.wikipedia.org/wiki/Strobilurin). We mentioned strobilurin as a synthetic commercial fungicide.

105: While this statement may be generally true, you are implying that this is true for oligomycins and there is no evidence that this is the case.

Our response:

Thank you for this comment. We have revised the statement.

131: “A pure culture of a single colony…” this is an odd statement; perhaps you meant, “A pure culture from a single spore.”

Our response:

Thank you for pointing out the error. We have revised the sentence.

136: The references cited, 14-29 do not refer to using 500 ml water to remove aerial mycelium.

Our response:

Thank you for this comment. We have used the technique of conidia production from the cited references 14 and 29. We have slightly optimized the method and used 500 ml water for removing aerial mycelia from the MoT agar block properly.

150: perhaps the ethyl acetate should have been a control as well.

Our response:

Thank you for the comment. We have used filter paper disks treated with ethyl acetate in the control plate, as we used water as a positive control in all the experiments.

150: You need to state the “active ingredient” (a.i.) of these materials. Are the oligomycines 100% pure? The fungicide is not; see https://www.cropscience.bayer.co.za/ en/Products/Fungicides/Nativo.aspx. I believe that the fungicide is 10 and 20% a.i. for the two components. 

Our response:

Thank you for this comment. We have used 100% pure powder form of oligomycins in our experiments. We purchased Nativo 75WG from Bayer Crop Science Ltd. Dhaka, Bangladesh and the a.i. for the two components is Trifloxystrobin/Tebuconazole 25:50 % w/w.

151: state the volume of material that was loaded into the disks

Our response:

Thank you for this comment. We have poured 20 µl compound solutions per disk for each concentrations.

155: you should state the space difference between the paper disc and the mycelial plug.

Our response:

Thank you for this comment. We have placed the paper disk at the opposite side of the mycelial plug in the 9 cm diameters petri plate. We have maintained about 2 cm distance between the paper disc and the mycelial plug.

156-158: Rather than call Nativo a positive control, I would refer to it as the “industry standard”. 

Our response:

Thank you for this comment. We have revised it. 

159: Earlier you referred to Nativo as a positive control and here you refer to it as a negative 

control.

Our response:

Thank you for the comment. We have used filter paper disk loaded with water as a negative control. We have revised the sentence.

163: This is unclear. I think you mean that each treatment, and the controls were done in duplicate and the experiment was repeated five times.

Our response:

Thank you for pointing out the editing error of this sentence. We have used five replications for each treatment for every experiment and repeated all the experiments five times. We have revised the sentence for clear understanding.

164: Did you measure the radius or the diameter; most people measure the diameter. I think the term radial is ok but it’s not clear what you measured.

Our response:

Thank you for the comment. We have measured the diameter of the fungal colony.

165: Since the colonies were not circular, you need to indicate where you took the measurements, in the middle of the colony? In the widest place? You measured in two directions, did you take the average? When I measure the growth in the figures in millimeters and did the math according to your equation, I came up with very different numbers. I probably measured differently than you.

Our response:

Thank you for the comment. We measured from two directions and took the average. We measured in centimeter with a meter ruler along with two perpendicular lines drawn on each plate's lower side.

170: To make these observations, did you make microscope mounts on slides? If from the petri dish did you observe through the top or the bottom of the dish? Did you use oil with the 100X lens?

Our response:

Thank you for the comment. We have observed through the bottom of the dish. We used oil immersion with the 100X lens.

170-172: “Hyphal morphology at the vicinity of compounds was observed…” and “…and images of morphological features of hyphae were recorded…” This needs to be clarified. I think you are saying. “Hyphal morphology at the leading edge of the colonies facing the treated and control discs were observed with a Zeiss Primo Star microscope at 40X and 100X. (you need to say how you did this, for example 100x is an oil emersion lens). Macro images of the disc diffusion experiment were captured with a Canon DOS 700D digital camera. Images of the hyphae were captured with a Zeiss Axiocam ERc 5s through the microscope.” I don’t know if I am describing your work correctly.

Our response:

Many thanks for describing our work appropriately. We have revised it.

184: Early you stated that washing the cultures was to remove aerial mycelium and here you say it was to reduce nutrients. And, the same references are used. I do not believe either reference is appropriate for this technique. Please re-read these references to be sure. Also, you need to be consistent as to why you washed the cultures. 

Our response:

Thank you for the query. For MoT conidia production, we used mycelial blocks from an untreated 5 days-old MoT fungus Petri plate and washed out the mycelia from the PDA agar blocks for washing out nutrient which is essential for conidia production according to this protocol. 

189: If you are going to say “optimum humidity and lights” then it implies that you know the humidity and light and you should express what it is; otherwise don’t mention it was optimum.

Our response:

Thank you for this comment. We have revised the sentence by adding the optimum humidity and lights (>90% RH and 14 h light followed by 10 h of darkness) requirement for sporulation according to our protocol.

210: Did you pipette the conidia onto a microscope slide and use a coverglass and oil-emersion? You need to explain exactly how you carried this out. The quality of the images in Figure 6 do not look like they were taken at 100X under oil. Also, it is troubling that the field of view shown varies considerably with regard to spore concentration. Spore germination can be affected by the concentration of spores. I realize the concentration was the same, but the Figure 6 shows few spores. It would have been better to selectively image the various stages (shown in Table 1) under oil with high quality images. 

Our response:

Thank you for the comment. We have pipetted the conidia onto a microscope slide and use a cover glass and oil-emersion. We used the same concentration of spores and we have provided representative images of various stages of conidia development. We have replaced some images according to the reviewer comment.

230: Describe the plates

Our response:

Thank you for the query. We have used sterile Sarstedt polystyrene square Petri plates (100 x 100 x 20 mm).

232: Did you inoculate 3 leaves for each treatment?

Our response:

Thank you for the query. We used 5 leaves for each treatment.

233: Was the leaf where the solution was placed, dry after 15 minutes?

Our response:

Thank you for the comment. We left the leaves to dry the solution for 15 minutes. 

234: 1 × 105 conidia/ml? or per 1 microliter?

Our response:

Thank you for the comment. We used ca. 1 × 105 conidia per 1 mililiter.

268: Why is there a reference to Dame et al.? Are the figures from a paper of his?

Our response:

Thank you for the comment. We have redrawn the chemical structures described in Dame et al. 

269: and Figure 2a: It appears that the inoculum and the control disc were placed next to each other; please explain.

Our response:

Thank you for the query. There were specific distance remains between the inoculum and the disk, but after the full growth of the fungus, it seems like they are placed next to each other.

308: I don’t see what is being described by “…destruction of regular growth with twisted ridges and corrugations, and irregular swelling of hyphal cells.” You must use some other descriptors; I can see differences but not twisted ridges and corrugations. Also, I do not see swelling of hyphal cells in the figures. Did you measure them?

Our response:

Thank you for the query. We observed under the light microscope at 100X magnification that oligomycins inhibited the regular circular hyphal growth of MoT by twisting and corrugating the ridges of hyphae. We also noticed irregular swelling of some hyphal cells. We have added images of swelling of hyphal cells in the figure taken at 100X magnification.

310: I don’t see the crystals in the Nativo figure, or the twisted ridges and corrugations. I am not sure I can see “loss of polar growth”. How does it grow without polar growth?

Our response:

Thank you for the query. We used the term ‘Crystal’ to describe the state of appearance of suppressed MoT hyphae by Nativo. Like oligomycins, Nativo also inhibited the hyphal growth of MoT by twisting and corrugating the ridges of hyphae. By the statement ‘loss of polar growth’, we meant that Nativo has arrested the normal polar growth of MoT hyphae. We have replaced the Figure 2(d) for better representation.

323: Figure 5. I can’t tell much from these figures. It is hard to justify them if they do not show what they need to show. Perhaps my copies are not very good.

Our response:

Thank you for pointing out this erroneous placement of the images in the figure. We have rearranged the images for better understanding. The images demonstrated the gradual decrease of sporulation with the increasing concentrations of compounds. Fig 5(a) represented huge sporulation in the control mycelial agar block, 5(b) represented gradual decrease in conidia production with increasing concentration of Oligomycin B; 5(c) and 5(d) represented gradual decrease in conidiogenesis with increasing concentration of Oligomycin F and Nativo. At 100 µg/ml, conidia production was zero in all the cases.

366: Do you mean that after the first 6 hours of germination, 14.3% had lysed? If so, I would write. “…and 14.3± 0.7% of the germ tubes lysed within the first 6 hr of incubation.”

Our response:

Many thanks for the correction. Yes, we mean it. We have revised it.

365 and 371 and Figure 6: Does “short” mean shorter than the control? Or does it mean normal? If it was shorter than the control, add “…shorter than the control”. You might add “approximately by half or a third”, or some other figure. Also, for figure 6, it would have been helpful to point out normal, short, long and appressoria in the figures. Unfortunately, the figures are of poor quality and do not show very well the differences. Since the counts and statistics were done on the observations, it was not necessary to take random pictures of the microscopic field; would have been better to selectively take images at higher magnifications illustrating the above differences in morphology.

Our response:

Many thanks for the corrections. It means shorter than control. We have revised the sentence. We have pointed out the normal, short, elongated and appressoria in the figures according to your suggestion. We have replaced some images for clear presentation.

379-380: “…but a slower growth of germ tubes…” This is the same thing as “short germ tubes”, right? If so, use the same descriptions for all three treatments.

Our response:

Thank you for pointing out the confusion generating sentence. We have revised the sentence.

382: It is not necessary to have (MoT) in the figure text since you will not refer to it again in the figure text.

Our response:

Thank you for the comment. We have revised it.

388-399: The subtitle as written is biased (as are some of the others in the results); it emphasizes the oligiomycins when the experiments are a comparison of a fungicide with the oligomycins.

Our response:

Thank you for this comment. We have revised it according to reviewer’s comment.

407 (Fig. 7) It needs to be indicated on the figure which leaves are represented by 5, 10 and 100 milligrams. You may also mention that “a representative leaf of five replicates”

Our response:

Thank you for the query. We have labelled below the images with each concentrations. We have revised it according to your suggestion.

423-427: Redundant sentences

Our response:

Thank you for the comment. We have deleted the sentences according to your suggestions.

435: It may be over-reaching to use the term “potential” without considering the possible economic cost of the application. Perhaps a bit of math may shed some light on this.

Our response:

Thank you for the comment. Here we used the term ‘potential’ to state about the effectiveness of the compounds to suppress MoT on detached wheat leaves.

439: Were these studies actually on the development of plant disease protection products?

Our response:

Thank you for this comment. We have reported some secondary metabolites against the wheat blast fungus to consider them as candidates for producing/developing commercial plant protection products (i.e., fungicides) with novel modes of action. 

444: I cannot see any evidence of swelling in Fig 2b and 2c. This should have been quantified by measurements and imaging of single hyphae. In the figure, where hyphae appear to be wider, it could easily be an artifact of the hyphae being close together. It should be a simple matter to treat some colonies, do the measurements and take new images of hyphae.

Our response:

Thank you for this query. We have found swelling of MoT hyphal cells under light microscope at 100X magnification, which we did not demonstrate. We have added the figures of swelling hyphal cells for better understanding.

445: this statement requires a reference or two.

Our response:

Thank you for the comment. We have addressed the statement with appropriate references.

446: what would have been the micrograms per ml?

Our response:

Thank you for the query. We have used a range of concentrations from 0.25 to 100 µg/ml. We loaded 20 µl of compounds solution in filter paper disk for each treatment. 

447: no data has been presented to show that increasing conc. Increases the dia of the hyphae

Our response:

Thank you for this comment. It was our observation but we did not present it in this paper. We have revised the sentence.

466: it is not clear that reduced conidiogenesis would occur on wheat leaves.

Our response:

Thank you for this query. We have given this statement on the basis of in vitro experiment. A further study is needed to confirm it on wheat leaves. 

472: you need to add arrows to the Fig to help this statement

Our response:

Thank you for the comment. We have added arrows in the Figure according to your suggestion.

482: No data was presented to support this statement. There may be a different was of bringing this idea forward.

Our response:

Thank you for this comment. We have deleted the statement, as it produced confusion.

---

## [Decision Letter · Decision Letter 1]

20 Jul 2020

PONE-D-20-13723R1

Oligomycins Inhibit Magnaporthe oryzae Triticum and suppress wheat blast disease

PLOS ONE

Dear Dr. Islam,

Thank you for submitting your manuscript to PLOS ONE. After careful consideration, we feel that it has merit but does not fully meet PLOS ONE’s publication criteria as it currently stands. Therefore, we invite you to submit a revised version of the manuscript that addresses the points raised during the review process.

The manuscript is much improved. Reviewer 1 could not critique the revised version of the paper, but my assessment is that their comments were adequately addressed. Reviewer 2 has additional comments and corrections to the manuscript that should be considered by the authors in a second round of revisions.

We look forward to receiving your revised manuscript.

Kind regards,

Richard A Wilson

Academic Editor

PLOS ONE

Reviewers' comments:

Reviewer's Responses to Questions

**Comments to the Author**

1. If the authors have adequately addressed your comments raised in a previous round of review and you feel that this manuscript is now acceptable for publication, you may indicate that here to bypass the “Comments to the Author” section, enter your conflict of interest statement in the “Confidential to Editor” section, and submit your "Accept" recommendation.

Reviewer #2: (No Response)

2. Is the manuscript technically sound, and do the data support the conclusions?

Reviewer #2: Partly

3. Has the statistical analysis been performed appropriately and rigorously? 

Reviewer #2: Yes

4. Have the authors made all data underlying the findings in their manuscript fully available?

Reviewer #2: Yes

5. Is the manuscript presented in an intelligible fashion and written in standard English?

Reviewer #2: Yes

6. Review Comments to the Author

Reviewer #2: (No Response)

7. PLOS authors have the option to publish the peer review history of their article (what does this mean?). If published, this will include your full peer review and any attached files.

Reviewer #2: No

---

## [Author Response · Author response to Decision Letter 1]

28 Jul 2020

Response to the Comments of Reviewers and Editor

Dear Editor,

We have revised the manuscript carefully addressing the queries/suggestions/comments of the reviewers, and uploaded a blue-inked marked version and a clean version. Here is our point by point responses to the reviewer’s comments.

Comments to the Author

1. If the authors have adequately addressed your comments raised in a previous round of review and you feel that this manuscript is now acceptable for publication, you may indicate that here to bypass the “Comments to the Author” section, enter your conflict of interest statement in the “Confidential to Editor” section, and submit your "Accept" recommendation.

Our response:

Many thanks for this encouraging comment.

Reviewer #2: (No Response)

2. Is the manuscript technically sound, and do the data support the conclusions?

Reviewer #2: Partly

Our response:

Thank you for your kind comment.

3. Has the statistical analysis been performed appropriately and rigorously? 

Reviewer #2: Yes

Our response:

Thank you for the encouraging comment.

4. Have the authors made all data underlying the findings in their manuscript fully available?

Reviewer #2: Yes

Our response:

Thank you for this encouraging comment.

5. Is the manuscript presented in an intelligible fashion and written in standard English?

Reviewer #2: Yes

Our response:

Thank you for the encouraging comment.

Comments by line

264-266: I think I know what you are saying here, unfortunately the images don’t show very well what you are describing. If possible, grow the fungus on these materials and then mount the hyphae on a glass slide and take images under oil at 100X. 

Our response:

Thank you for the comment. For your kind observation and understanding, we have enclosed below two images of swelling of MoT hyphal cells under the effect of oligomycin B and F that were taken at 100X under oil, for your kind consideration. Due to COVID-19, our university and laboratories are under lockdown, and hence, it is difficult for us to conduct further experiment in this pandemic condition.

I will try to write this sentence differently, but I don’t want to change the meaning. 

“Hyphae treated with oligomycin B and F showed irregular growth and an increase in branch frequency per unit length of hyphae. The hyphal cell walls were not smooth but showed ridges giving a corrugated appearance and irregular swelling of cells (Fig 2b and 2c).”

Our response:

Thank you for this valuable suggestion for the improvement of the manuscript. We have revised the text accordingly.

268: you do not need to repeat the description of the hyphae since you said it was essentially the same. I do not know what you mean by crystals, but it is probably the wrong word. I cannot help you with that because I don’t see anything that resembles crystals in the figure.

Our response:

We agree with your valuable suggestion. Thank you. We have revised it according to your suggestion. We have deleted the confusing term ‘crystal’ for clarification.

270: The only difference I see in the description of the hyphae are “crystals”. Did crystals develop in the culture medium, or the hyphae? 

Our response:

Thank you for this critical and important comment. There was no real crystals developed in the culture medium, or the hyphae. We used the term ‘Crystal’ to describe the state of appearance of suppressed MoT hyphae by Nativo. However, we have deleted this confusing/inappropriate term ‘crystal’ and revised the manuscript accordingly.

274-282: This paragraph is very awkward. And I cannot tell much by the pictures. I gather that the dark areas are clusters of conidia. The 5 microgram images appear out of focus. There seems to be more conidia in the 5 microgram images which is a good thing. At 100 micrograms there is a lot of hyphae which does not show up in the other images, why is that? Is the picture taken at a different plane of focus? That would explain the differences. You would not expect much hyphae growth in 24 hr. so that must be background hyphae. It would have been better not to have focused on the hyphae because it is distracting and a bit misleading. The control does not appear to have hyphae either. You need to discuss the differences in these images.

Our response:

Thank you for pointing out the weakness of the paragraph. We stated before that Figure 4 represents the gradual decrease of conidia production with the increasing concentrations of the tested compounds. Yes, the dark areas are the clusters of conidia. We have replaced the 5 microgram image of oligomycin F with a clear one for better representation.

The 100 micrograms images represent the existence of broken mycelial tips with no sign of conidiophore and conidial growth on the surface of the agar block treated with macrolides and Nativo. There was not any hyphal growth occurred after the treatment of macrolides and Nativo at 100 µg/ml. Yes, we have taken the picture in slightly different plane of focus as the conidia produces on the conidiophore on the upper side of the hyphae. However, we have replaced the images with same plane of focus for better representation.

I assume (and the images in Fig 4 seem to show it) that for both the oligomycins and the fungicide, there is inhibition correlated with dose. If this is true, then maybe the following paragraph will be clearer. I am assuming that conidiogenesis and formation of conidia mean the same thing.

Inhibition of conidiogenesis

Conidial formation by MoT was remarkable decreased by the oligomycins and the fungicide at 5 and 10 µg/ml when compared to the control, and inhibition increased with an increase in concentration from 5, 10 and 100 µg/ml (Fig 4). For all three treatments, no conidia developed at 100 µg/ml. Microscopic examination revealed burst mycelial tips, and a complete lack of conidiophores for all three treatments at 100 µg/ml.

Our response:

Thank you for kind and valuable suggestions for the improvement of the manuscript. We have revised the paragraph accordingly. 

315: the sentence as written is a little unclear. Since there was no germination, obviously there could not be germ tubes or mycelial growth. Simplify.

“It is interesting that the oligomycins produced abnormally long or short germ tubes and branching and lysing while the fungicide did not result in these changes.”

Our response:

Thank you for your kind suggestion. We have revised it as per your suggestion.

350-352: based on Fig. 6A the opposite appears to be true. I would write “At 5 µg/ml the fungicide suppressed lesion development more than both oligomycins. At 10 µg/ml the fungicide showed more suppression than oligomycin F. No lesions developed with any of the three treatments at 100 µg/ml.

Our response:

Thank you for correcting and revising the sentences. We have revised these sentences.

383: How was o-F 10-fold stronger than Nativo? How was the calculation made? Fig. 6A shows the fungicide more effective (shorter lesions) than o-F.

Our response:

Thank you for the query. In bioassay, we have found that oligomycin F was 10-fold stronger than Nativo in terms of mycelial growth inhibition, where the minimum inhibitory concentration of oligomycin F and Nativo were 0.005 and 0.05 μg/disk, respectively. We have revised the sentence for better understanding.

409: Not sure of the relevancy of this statement. For this kind of pathogen you have spore germination, germ tube, appressorium, infection and then hyphae development the host.

Our response:

Thank you for the comment. 

452-454: scale was not mentioned or defined in the text so hear it is confusing. I would delete this section as it is redundant.

Our response:

Thank you for the comment. We have revised it according to your suggestion.

458-459: this is not what Fig 6B shows

Our response:

Thank you for pointing out an error. We have revised it.

481-484: this is an overstatement and should be deleted.

Our response:

Thank you for the kind suggestion. We have deleted the statement.

We sincerely appreciate the knowledgeable reviewer for valuable editorial comments for the improvement of the manuscript.

A revised version mentioning the changes in blue-ink and a clean version were uploaded for the find perusal and consideration.

We hope that this revised version (R2) will be accepted for publication in the PLOS ONE.

Kind regards,

Dr. Md Tofazzal Islam, FBAS 

(Fellow of Fulbright, Commonwealth, JSPS and Alexander von Humboldt Foundation) 

Professor and Director, Institute of Biotechnology and Genetic Engineering (IBGE),

Bangabandhu Sheikh Mujibur Rahman Agricultural University, Gazipur-1706

BANGLADESH

Tel. +88-02-9205310-14 Extn. 2252

Fax: +88-02-9205333

Cell: +88-0171-4001414, +88-01534568893

http://www.btlbsmrau.org

http://www.researchgate.net/profile/Md_Tofazzal_Islam

http://orcid.org/0000-0002-7613-0261

---

## [Editor Report · Decision Letter 2]

30 Jul 2020

Oligomycins Inhibit Magnaporthe oryzae Triticum and suppress wheat blast disease

PONE-D-20-13723R2

Dear Dr. Islam,

We’re pleased to inform you that your manuscript has been judged scientifically suitable for publication and will be formally accepted for publication once it meets all outstanding technical requirements.

Kind regards,

Richard A Wilson

Academic Editor

PLOS ONE
---

## [Editor Report · Acceptance letter]

3 Aug 2020

PONE-D-20-13723R2 

Oligomycins Inhibit Magnaporthe oryzae Triticum and suppress wheat blast disease 

Dear Dr. Islam:

I'm pleased to inform you that your manuscript has been deemed suitable for publication in PLOS ONE. Congratulations! Your manuscript is now with our production department. 

Kind regards, 

on behalf of

Dr. Richard A Wilson 

Academic Editor

PLOS ONE